# Transreplication Preference of the Tomato Leaf Curl Joydebpur Virus for a Noncognate Betasatellite through *Iteron* Resemblance on *Nicotiana bethamiana*

**DOI:** 10.3390/microorganisms11122907

**Published:** 2023-12-01

**Authors:** Thuy T. B. Vo, I Gusti Ngurah Prabu Wira Sanjaya, Eui-Joon Kil, Aamir Lal, Phuong T. Ho, Bupi Nattanong, Marjia Tabassum, Muhammad Amir Qureshi, Taek-Kyun Lee, Sukchan Lee

**Affiliations:** 1Department of Integrative Biotechnology, Sungkyunkwan University, Suwon 16419, Republic of Korea; bichthuy251188@gmail.com (T.T.B.V.); gusti.prabu20@gmail.com (I.G.N.P.W.S.); hophuongk59sinhhoc@gmail.com (P.T.H.); gum.bupi@gmail.com (B.N.); marjia39@g.skku.edu (M.T.); amirq303@gmail.com (M.A.Q.); 2Department of Plant Medicals, Andong National University, Andong 36729, Republic of Korea; viruskil@anu.ac.kr (E.-J.K.); aamirchaudhary43@gmail.com (A.L.); 3Ecological Risk Research Department, Korea Institute of Ocean Science & Technology, Geoje 53201, Republic of Korea

**Keywords:** transreplication, non-cognate betasatellite, tomato leaf curl Joydebpur virus, infectious clone construction, *iteron*

## Abstract

Pepper plants (*Capsicum annuum*) with severe leaf curl symptoms were collected in 2013 from Bangalore, Karnataka, India. The detection results showed a co-infection between the tomato leaf curl Joydebpur virus (ToLCJoV) and tomato leaf curl Bangladesh betasatellite (ToLCBDB) through the sequencing analysis of PCR amplicons. To pinpoint the molecular mechanism of this uncommon combination, infectious clones of ToLCJoV and two different betasatellites—ToLCBDB and tomato leaf curl Joydebpur betasatellite (ToLCJoB)—were constructed and tested for their infectivity in *Nicotiana benthamiana*. Together, we conducted various combined agroinoculation studies to compare the interaction of ToLCJoV with non-cognate and cognate betasatellites. The natural non-cognate interaction between ToLCJoV and ToLCBDB showed severe symptoms compared to the mild symptoms of a cognate combination (ToLCJoV × ToLCJoB) in infected plants. A sequence comparison among betasatellites and their helper virus wasperformed and the *iteron* resemblances in ToLCBDB as well as ToLCJoB clones were processed. Mutant betasatellites that comprised *iteron* modifications revealed that changes in *iteron* sequences could disturb the transreplication process between betasatellites and their helper virus. Our study might provide an important consideration for determining the efficiency of transreplication activity between betasatellites and their helper virus.

## 1. Introduction

Tomato leaf curl Joydebpur virus (ToLCJoV) is a monopartite begomovirus belonging to the *Geminiviridae* family and poses a major threat to tomato productivity in the eastern part of India. This virus was first reported in Bangladesh in 2005 and its infection spreadto India later [1,2,3]. The ToLCJoV genome comprises a 2.7 kb single-stranded DNA (ssDNA) and is analogous to the DNA A component of a bipartite begomovirus. The genome contains six open reading frames (ORFs) in both viral- and complementary-sense strands [4]. The viral-sense strand presents the precoat protein (V2) and coat protein (V1), whereas the complementary-sense strand encodes the replication initiation protein (Rep, C1), replication enhancer protein (Ren, C3), transcriptional activator protein (TrAP, C2), and C4 protein [5].

Monopartite begomoviruses are usually found to be associated with DNA satellites. This association may change the host range or increase the pathogenicity of the virus [6]. There are three types of DNA satellites associated with begomoviruses. Alphasatellites consist of circular ssDNA molecules with half the size of the begomovirus component and encode a replication-associated protein (Rep) that allows this satellite to replicate autonomously [7]. Betasatellites with half the size of the begomovirus component encode a βC1 protein that has important roles in symptom induction and the suppression of transcriptional and post-transcriptional gene silencing [8,9]. Deltasatellites, about a quarter the size of the begomoviral genome component, are noncoding DNA satellites associated with begomoviruses [5].

In our first study, ToLCJoV was detected with a non-cognate satellite (ToLCBDB) instead of being associated with its cognate betasatellite in symptomatic pepper plants from Bangalore, India. This finding raised a question relating to the transreplication activity between the monopartite begomoviruses represented by ToLCJoV and their satellite molecules. Betasatellites are extra viral components that do not encode CP but can influence the pathogenesis and accumulation of the associated helper viruses and are considered part of the begomovirus complex [10]. In most cases, cognate helper viruses replicate satellites to higher levels than non-cognate viruses [11,12,13], as previous research has suggested that a replication specificity exists between the begomovirus and betasatellite [14]. Betasatellites cannot replicate themselves and are completely dependent on the helper virus for replication, encapsidation, movement, and insect transmission [15]. The replication of betasatellites is mediated by the Rep-associated protein of the helper virus, where it initiates viral DNA replication by binding to reiterated motifs (*iterons*) present in the common region (CR) and introducing a nick into the conserved nonanucleotide TAATATTAC [5]. Some previous studies reported that a helper begomovirus could transreplicate two different betasatellites [13,16,17]; for instance, the tomato yellow leaf curl China virus and tobacco curly shoot virus can transreplicate and stably maintain either a tomato yellow leaf curl China betasatellite or tobacco curly shoot betasatellite. However, when both satellites molecules are co-inoculated with each helper virus, only the cognate betasatellite can be stably maintained in the later stage of the plant [14]. Also, several other mutations have been studied and have demonstrated the necessity of virus-encoded genes for the transreplication and maintenance of betasatellites [18,19].

The mechanism of how a betasatellite can be transreplicated by its helper virus is still poorly known. To improve the understanding of the interaction between betasatellites and their helper virus, two combinations of ToLCJoV × ToLCBDB and ToLCJoV × ToLCJoB were compared using infectious clones via agro-inoculation in this study. The infectivity assay resulted in differences in symptom development in *N. benthamiana*. For an in-depth study in transreplication activity, we focused on several mutant betasatellites of ToLCBDB and ToLCJoB based on *iteron* modification. Their infectious clones were generated and challenged in tobacco plants. The results demonstrated the role of *iterons* in the infectivity of ToLCJoV with cognate and non-cognate betasatellites. Our study provides a greater knowledge of thetransreplication mechanism between betasatellites and their helper virus.

## 2. Materials and Methods

### 2.1. Samples Collection and Viral Detection

Sixteen pepper plant samples with severe leaf curl symptom were observed and collected from one commercial field in Bangalore, India (12.9716° N, 77.5946° E). The sodium-Tris-EDTA (STE) method [20] was used to perform all DNA extractions in this study using STE buffer (EDTA (20 mM; pH 8), Tris-HCl (2M; pH 8), and 4M sucrose). Total DNA was used for PCR amplification using primer sets which were previously published [21,22], to initially identify the virus in the collected samples. The PCR products were sequenced using the commercial service Macrogen (Seoul, Republic of Korea). Also, the total DNA was a template for rolling circle amplification (RCA) using the TempliPhi™ 100 Amplification kits (Cytiva, Marlborough, MA, USA) following the manufacturer’s instructions to obtain a full-length genome sequence for the virus. The RCA products were digested with different enzymes and loaded in 1% agarose gel. To identify the putative full length sequence, the resulting fragments were cloned into the pGEM^®^-T Easy vector (Promega, Madison, WI, USA) and sequenced by commercial service Macrogen (Seoul, Republic of Korea). The recombinant plasmids were sequenced and submitted to the GenBank database. The full-length sequences were examined using CLC Sequence Viewer version 8.0 software (https://clc-sequenceviewer.software.informer.com/8.0/, accessed on 8 November 2022). Also, the putative recombination of isolated ToLCBDB was analyzed using different recombination detection methods implemented on the RDP4 v4.100 suite [23].

### 2.2. Phylogenetic Tree and Sequence Demarcation Tool (SDT) Analysis

The phylogenetic tree was constructed using the Tamura-Nei model and the neighbor-joining method with 1000 bootstrap steps using MEGA-X [24,25]. Different full-length sequences of ToLCJoV identified in various countries were obtained fromGenBank and aligned using BLAST in MEGA-X. The Tamura-Nei model and neighbor-joining method, along with 1000 bootstrap steps, were used to display the tree graph. Additionally, a color-coded pairwise identity matrix was generated from the same sets of sequences using the program Sdt v1.2 [26]. Each colored cell represents a percentage identity score for two sequences.

### 2.3. Infectious Clone Construction

Infectious clones of ToLCJoV, ToLCBB and ToLCJoB were constructed usingthe partial tandem repeat method as previously described [27]. To construct the ToLCJoV infectious clone, each partial fragment of approximately 1.6 kb and 1.5 kb was amplified by the newly designed primer sets based on its full genome (MZ605914 obtained from symptomatic chili plants) (Appendix A) via PCR amplification. Then the amplified fragments were ligated into a pGEM^®^-T Easy vector and transformed into *Escherichia coli* strain DH5α using the heat shock method [28] to generate pGEM-ToLCJoV-0.6 mer/−0.5 mer. The transformed plasmids from *E. coli* were digested by three restriction enzymes *Xho*I, *Pst*I and *Bgl*II and then ligated into the pCAMBIA 1303 vector to obtain the pCAMBIA-ToLCJoV-1.1.mer construct. The recombinant pCAM-ToLCJoV-1.1 mer was transformed into the *Agrobacterium tumefaciens* strain GV3101 through electroporation. Using the same approach ToLCBDB and ToLCJoB infectious clones were constructed based on the full-length genome (ToLCBDB: MZ605915 and ToLCJoB: NC010236). *Hind*III, *Sac*I, and *Spe*I were used for ToLCBDB while the ToLCJoB clone required *Hind*III, *Sma*I, and *Spe*I to generate pCAMBIA-ToLCBDB-2 mer and pCAMBIA-ToLCJoB- 2 mer constructs.

### 2.4. Agroinoculation Assay

*N. benthamiana* seeds were sown in sterilized soil and cultivated in a growth chamber at Sungkyunkwan University, Suwon, Republic of Korea. Three-week-old plants with similar sizes were selected to inoculate the infectious clones and five plants were inoculated with Agrobacteria containing empty pCAMBIA 1303 as mock plants. *Agrobacterium* harboring recombinant plasmid of each clone was cultured in a 28 °C shaking incubator at 250 rpm for 40 h until OD at 600 nm reached 1.0. Inoculation was performed using the pin-pricking method using an insect pin [29]. Inoculated plants were grown and observed for symptoms in a non whitefly plant growth chamber (a photoperiod of 16 h light and a target air temperature set at 28/22 °C day/night).

### 2.5. Combination of Inoculation and Satellites Mutant Construction

To perform a comparison of symptom development and transreplication preferences, three combinations of inoculation were prepared: ToLCJoV only, ToLCJoV × ToLCJoB, and ToLCJoV × ToLCBDB. Each combination was inoculated into five plants of *N. benthamiana*.

Assuming that the *iteron* sequence plays a role in the transreplication process between betasatellites and their helper virus, three mutant viruses were constructed: ToLCBDB mutant (ToLCBDB M1), ToLCJoB mutant 1 (ToLCJoB M1), and ToLCJoB mutant 2 (ToLCJoB M2).These mutant viruses were made by constructing nucleotide modifications in the *iteron* part, while others were the same as the wild-type clones. In particular, the “GGAGTC” (before the TATA box) and “GGTATC” sequences (after the TATA box) of ToLCBDB were mutated to “GGTGTC” and “GACACC”, respectively, to obtain the ToLCBDB M1 clone. Simultaneously, to obtainToLCJoB M1, two sequences including “GATACC” and “GGCACC” in the TATA box upstream were changed to “GACACC”, while “GGTGCC” was replaced by “GGTGTC” placed downstream of the TATA box. The ToLCJoB M2 contained three “GGTGTC” and one “GACACC” in the same position as the wild-type ToLCJoB sequence. These mutant sequences were synthesized by Macrogen (Seoul, Republic of Korea) and cloned into the pCAMBIA 1303 vector to achieve the mutant infectious clones.

### 2.6. Viral DNA Detection and Southern Hybridization Blot

Young leaves of each group were harvested at the fourth week post inoculation. The collected samples were extracted using the STE extraction method, as previouslydescribed. PCR was performed with the specific detection primer set for each virus (Table 1). The PCR reaction was conducted in a T100 thermal cycler (Bio-Rad, Hercules, CA, USA) using a mixture with the following reagents, 10 µL of 1× *AccuPower*^®^ PCR Mastermix (Bioneer, Oakland, CA, USA), 1 µL of template DNA, 1 µL of primer forward (10 pM), 1 µL of primer reverse (10 pM), and distilled water, to reach a total volume of 20 µL. The ToLCJoV PCR condition was as follows: preheating at 94 °C (3 min), 35 cycles of denaturation at 94 °C (30 s), annealing at 58 °C (30 s), extension at 72 °C (1 min), followed by a final extension at 72 °C (5 min). Meanwhile, ToLCBDB and ToLCJoB PCR conditions were only different at annealing temperature, 55 °C (30 s). Amplified DNA was analyzed using electrophoresis on 1% agarose gels, stained with ethidium bromide for the target size, and captured using the Digital gel documentation model GDS-200D (Korea Lab Tech, Seong Nam, Republic of Korea).

Southern blot hybridization was performed to identify viral DNA replication from experimental plant samples [30]. Briefly, the extracted DNA (20 µg) was mixed with a 10× loading buffer (Takara, Otsu, Shiga, Japan) and loaded onto a 1% agarose gel, and then separated through electrophoresis at 30 V for 8 h. After electrophoresis, the gel was depurinated in 0.2 N HCl for 10 min, denatured in 0.4 M NaOH for 15 min, and neutralized in 0.5 M NaCl solution for 30 min. The DNA was transferred to a positively charged nylon membrane (Hybond-N^+^ membrane, GE Healthcare, UK) using the capillary transfer method for 16 h. After transfer, the nylon membrane was exposed to ultraviolet radiation for 2 min using an ultraviolet crosslinker machine (UVC 500 crosslinker, GE Healthcare) to permanently attach the transferred DNA to the membrane. The PCR product for the coat protein-coding sequence of ToLCJoV, for the partial βC1 gene of ToLCJoB and ToLCBDB via detection primer (Table 1), was labeled with [α-^32^P] and used as a probe. Hybridization was performed at 65 °C for 12 h in a hybridization buffer. The nylon membrane was washed with 2× SSC (saline-sodium citrate), 0.1% SDS, and 1× SSC + 0.1% SDS buffer each for 30 min. After washing, the membrane was exposed to X-ray film (Agfa-Gevaert N.V., Mortsel, Belgium) for at least 24 h in a −80 °C deep freezer.

### 2.7. Quantitative PCR to Determine Viral Accumulation

Quantitative PCR (qPCR) was performed on agroinoculated plants at 28 days post inoculation (dpi) to determine viral accumulationusing our previously established protocol [31]. In brief, equal 20 ng of different genomic DNA sampleswas used as templates in the qPCR reaction, which included 5 µL of TBGreen^®^ Premix Ex Taq™ II (TliRNaseH Plus; TaKaRa Bio, Kusatsu, Shiga, Japan), 1 µL of 10 pMof each qPCR primer set (Table 1), and sterile water to a final volume of 15 µL. EF-1α utilized as the internal control. The reaction conditions were as follows: pre-denaturation at 95 °C for 10 min followed by 35 cycles (for ToLCJoV) and 40 cycles (for ToLCJoB) of a denaturation step at 95 °C for 10 s, an annealing step at 58 °C (ToLCJoV) and 60 °C (ToLCJoB) for 15 s and an extension step at 72 °C for 20 s. Data analysis was performed using the 2^−ΔΔCt^ method [32]. Statistical analyses were conducted using the t-test using GraphPad Prism version 8.0 (GraphPad Software, La Jolla, CA, USA).

## 3. Results

### 3.1. Virus Detection and Sequence Analysis via Phylogentic Tree and SDT

Symptomatic pepper plants were collected from Bangalore, Karnataka, India (Figure 1A,B). Total DNA was extracted and used as a template to perform PCR with begomovirus universal primers. Sequencing analysis of the obtained 1 kb amplicon showed a high similarity with ToLCJoV. Additionally, RCA was conducted to obtain the full-length ToLCJoV genome in the collected samples. Among several restriction enzymes used to digest RCA products, only *Bam*HI showed some putative bands. Of these, six samples showed two bands of 2.7 and 1.3 kb band had a similar size to that of the satellite molecules. The results of this study showed that the begomoviruswas associated with their betasatellite in the plant samples. To identify them, a full-length construction of the two associated bands was created through multiple cloning steps and sequencing analysis. The sequence analysis of the separately amplified fragments revealed that the 2.7 kb sequence was ToLCJoV (accession number MZ605914) with 98.41% identical to EU431116, which was isolated from *Hibiscus cannabinus* in India [2]. Similar to other monopartite begomoviruses, this isolate of ToLCJoV contains six ORFs (Appendix A). The intergenic region (IR) was 350 nt, containing a highly conserved sequence forming a stem-loop structure, TAATATT↓AC, found in most geminiviruses. The *iteron* sequence also confirmed the presence of the nonanucleotide sequence GGTGTC (2625–2630 nt, repeated at 2653–2658 and 2660–2665 nt).The other 1.3 kb band was identified as ToLCBDB (accession number MZ605915) and showed a 98.47% identity to HM143910. It containsa single ORF βC1 (201–563 nt) encoding the 13.818 kDa protein (Appendix A). The conserved nonanucleotide sequence TAATATT↓AC and *iteron* sequences GGAGTC (1072–1078 nt), GGTACC (1095–1100 nt), and GGCACC (1107–1112 nt)were also confirmed.However, these *iteron* sequences were not completely repeated but only had some identical nucleotide patterns. A comparison of the sequence of ToLCJoV in this study to other reported ToLCJoVs associated with betasatellites revealed an identity of around 91% to 98% of this ToLCJoV with the reported cases. Among all ORFs, there were no significant difference between isolates, but the IR part of ToLCJoV in this study was the most diverse, with an average identity of 84% to 97% compared to the other isolates (Appendix A).

Interestingly, in the samples, it was found that ToLCJoVwasassociated with ToLCBDB rather than ToLCJoB-the cognate betasatellite for ToLCJoV. To determine the sequence relationship between the two betasatellites, a phylogenetic and SDT analysis was conducted. TheToLCBDB isolated in this study and ToLCJoB formed a different clade on the phylogenetic tree. Our ToLCBDB is closely related to ToLCBDB isolated from chilli in India. The data also showed that ToLCJoB shares the same clade with other ToLCJoBs isolated from chilli (Figure 2A). Because the IR is one of the important factors in begomovirus replication, there is potential to find its relation as a clue toward the transreplication preference between these viruses [33]. Thus, the IRs of ToLCJoV, ToLCBDB, and ToLCJoB were also aligned and identity percentages were calculated using the SDT program (Figure 2B). Coinciding with phylogenetic data, the IR of ToLCBDB isolated in our study has the highest identity with ToCLBDB from India (>95%), while ToLCJoB showed an almost 100% similarity with other Indian ToLCJoV isolates. These results indicated that the ToLCBDB isolated in our study is a variant of the Indian-reported ToLCBDB, also genetically distant fromToLCJoB. Also, the recombinant analysis detected recombination events in the IR of ToLCBDB. The recombinant nucleotide coordinates are 658–900, likely resulting from a recombinant event of the ToLCJoV IR (27.2%) according to RDP analysis, with a *p*-value of 1.61 × 10^−19^.

### 3.2. Infectivity of ToLCJoV with Different Betasatellite

The infectivity and symptom development of this virus combination were assessed throughagroinoculation in *N. benthamiana* plants. Clear differences in symptom development were observed between each combination. For ToLCJoV × ToLCBDB combination, leaf curling and leaf crumpling symptoms were induced in *N. benthamiana*, while the ToLCJoV × ToLCJoB showed mild leaf curling symptoms (Figure 3A, Table 2). Thissuggestes that ToLCBDB may havetransreplicated and simultaneously altered the disease progression for ToLCJoV leading to severe symptom induction, whereas ToLCJoB failed to exhibit severe pathogenicity in *N. benthamiana* with an unknown mechanism. The infectivity of each combination was also confirmed at 4weeksst inoculation (wpi) through PCR results. The specific amplicon was detected in the PCR product of all combinations, even though ToLCJoB showed a light band (Figure 3B). Additionally, qPCR data indicated differences in ToLCJoV accumulation among each combination (Figure 3C). ToLCJoV accumulation was highest in the ToLCJoV × ToLCBDB condition compared to others. Furthermore, Southern blot hybridization results confirmed viral replication with the presence of supercoiled and open circular forms of dsDNAtogether with an ssDNA band (Figure 3D). These results indicate that all three clones are infectious and also present differences in disease phenotype when co-inoculated with ToLCJoV with cognate and non-cognate betasatellites.

### 3.3. Infectivity of Mutant Clones

Our study focused on the interaction between betasatellites and their helper virus, specially examining the DNA motif known as the *iteron*. A previous study on thetransreplication in begomovirusessuggested that certain DNA motifs could play a role in selective replication between betasatellites and their helper virus [13]. We conducted an in silico analysis to compare the *iteron* sequences of ToLCJoV, ToLCJoB, and ToLCBDB. To further investigate the function of *iterons* in transreplication between ToLCJoV and its cognate/non-cognate betasatellite, we constructed three mutants with modified *iteron* sequences in ToLCBDB and ToLCJoV. These mutant betasatellites, ToLCBDB M1, ToLCJoB M1, and ToLCJoB M2, were designed to have the same *iteron* sequences as ToLCJoV in either the forward or reverse direction (Figure 4).

Further investigation into the correlation between iteron resemblances and viral pathogenicity was conducted through agroinoculation using different combinations: ToLCJoV × ToLCJoB M1, ToLCJoV × ToLCJoB M2, and ToLCJoV × ToLCBDB M1 (Figure 5A). Based on *iteron* modification, betasatellites mostly failed to replicate with the assistance of ToLCJoV as their helper virus. The leaves of ToLCJoB M1-infected plants appeared more crumpled than the other mutants that exhibited the same mild crumpling as the phenotype of ToLCJoV infection alone (Figure 5B). DNA extraction and PCR detection results showed that only ToLCJoB M1 could still be replicated (Figure 5C, Table 2), but the other mutant betasatellites could not be detected in all inoculated plants. Among all mutant betasatellites, ToLCJoB M1 has the smallest changes, with only three nucleotides different from its *iteron* sequence compared to the original virus. We also assessed the relative viral accumulation in infected plants using the qPCR reaction as described in the Materials and Methods section. According to the results, the accumulation of ToLCJoV and betasatelliteDNA in ToLCJoB M1-infected plants was higher compared to others (Figure 5D). Additionally, strand-specific amplification was performed using virion-sense and complementary-sense specific primer sets based on previous reports [34]. The results revealed that the infected plants had both dsDNA and two ssDNA molecules (virion and complementary senses), indicating that the viral genomes replicated normally in *N. benthamiana*. These results suggest that the *iteron* may play an important role in affecting the infectivity of the betasatellite in combination with the helper virus.

## 4. Discussion

Monopartite begomoviruses are often associated with betasatellites, which can enhance the pathogenicity of the begomovirus and enable it to infect hosts outside of its natural host range. Betasatellites rely on their helper virus for transreplication activity to maintain their presence. However, not all begomoviruses can transreplicate other betasatellites, and the specific mechanism of transreplicationremains uncharacterized. Previous studies have indicated that the Rep binding site isa crucial factorin determining the efficiency of replication and transreplication of begomoviruses [14,35]. Geminiviruses replicate using a rolling-circle mechanism initiated through the Rep binding to specific repeated sequence motifs called *iterons* upstream of the stem loop (TAATATTAC) in the IR [36]. Additionally, in the presence ofbetasatellites, the interaction between the helper virus’s Rep and specificmotifs in the betasatellite IR is considered an important step in betasatellitetransreplication.

Our study detected a natural association between ToLCJoV and ToLCBDB-the non-cognate betasatellite-resulting in severe disease symptoms in pep-per plants that were collected from the open field. To confirm the field observation as well as prepare for further studies, the infectious clones of ToLCJoV and two betasatellites, ToLCBDB and ToLCJoB, were constructed and tested in N. benthamiana with different combinations. In the comparison of infectivity, the ToLCJoV and ToLCBDB combination showed the severe disease phenotype in N. benthamiana compared to the mild symptoms of the ToLCJoV and ToLCJoB combination. This result indicated that ToLCJoV and ToLCBDB have specific interaction preferences compared to ToLCJoV and ToLCJoB as a cognate as-sociation. An investigation to reveal this transreplication activity preference was con-ducted through in silico analysis, specifically in the Rep binding site sequence (iteron) in-side the IR of both betasatellites. The result from in silico analysis revealed a stronger iter-on resemblance between ToLCJoV and ToLCBDB than ToLCJoV and ToLCJoB. This could be one of the possible reasons for the transreplication preference between ToLCJoV and ToLCBDB.

Further, our study showed the importance and effects of *iteron* on the transreplication activity of betasatellites and their helper virus. Three mutant betasatellites with modifications in the *iteron* sequence were constructed and challenged in *N. benthamiana* plants. Results of mutant betasatellite inoculation showed that only ToLCJoB M1 can be transreplicatedat the same level as the wildtype. Among all mutant betasatellites, ToLCJoB M1 had the smallest changes, withonly three nucleotides different from its *iteron* sequence compared to the wild-type virus. These small changes make the *iteron* sequence of ToLCJoB not too different from the wildtype; hence, it can still be recognized by Rep of ToLCJoV. The transreplication activity between ToLCJoV and ToLCBDBissurely involved in the *iteron* relation of these two viruses if we refer to the *iteron* function as a specific Rep binding site that starts the replication process. The resemblance of *iterons* between betasatellites and their helper virus can be considered as one of the important factors determining the efficiency of transreplication activity. In our case, the resemblance of *iterons* is not completely the same, because the ToLCJoV *iteron* sequence (GGTGTC, GGTGTC, GGTCTC, GACACC) is not identical to ToLCBDB (GGAGTC, GGTACC, GGCACC, GGTATC). *Iteron* recognition does not need the whole *iteron* sequence, but the first two or three bases might be more important in the recognition phase by Rep [36].However, it is still unclear how Rep mediates origin recognition and *trans* replication of betasatellite DNA. According to our results, iteron resemblance is not the only key factor determining the success of cognate and non-cognate betasatellite iterons in transreplication activity, but the motif orientation also plays an important role in it. For the cognate betasatellite, the mu-tant ToLCJoB M2 with the same iteron sequence as the helper virus failed to transreplicate. By contrast, ToLCJoB M1 did not resemble the ToLCJoV iteron completely but maintained the same transreplication efficiency as the wild-type one, suggesting that high-affinity Rep replication origin binding (GGTGTC) is required in the inverted orientation for ToLCJoB replication, which is different from the previous report about the binding site requirement in the cognate satellite of ToLCV [37]. This difference may be caused by unknown factors in the replication process among different viruses. In the case of the non-cognate betasatel-lite, the results from our study revealed that the direction of the TATA box’s motif down-stream may be important when our mutant in that motif (ToLCBDB M1) abolishes the sat-ellite replication. Additionally, our studies suggested that the high-affinity binding site (GGTGTC) is not specific for ToLCBDB replication, and the binding site is not identical to ToLCJoV iterons. This could be the possibility to determine the difference in the interaction between the helper virus and non-cognate betasatellite, compared to the cognate one, which requires the iteron homologous with the helper virus in its replication [13]. Even though many further studies are still required, our finding showed that iteron sequence direction might be an important consideration for determining the efficiency of transrep-lication activity between betasatellites and their helper virus.

## 5. Conclusions

Our study focused on the interaction between ToLCJoV and the cognate betasatellite ToLCJoB or non-cognate betasatellite ToLCBDB. The results demonstrate that the infection of pep-per-isolated ToLCJoV was ameliorated in the presence of non-cognate ToLCBDB as com-pared to its cognate satellite ToLCJoB. Moreover, the important role of iterons has been con-firmed through many infectivity assays with mutant infectious clones. Although further studies are required, our study has generated information about the dynamics of co-infection of monopartite begomovirus and non-cognate betasatellite. We also high-lighted the role of viral iteron sequences between the viral helper DNA and satellite mole-cules and provided helpful information for the researchers working on geminiviruses.

## Figures and Tables

**Figure 1 microorganisms-11-02907-f001:**
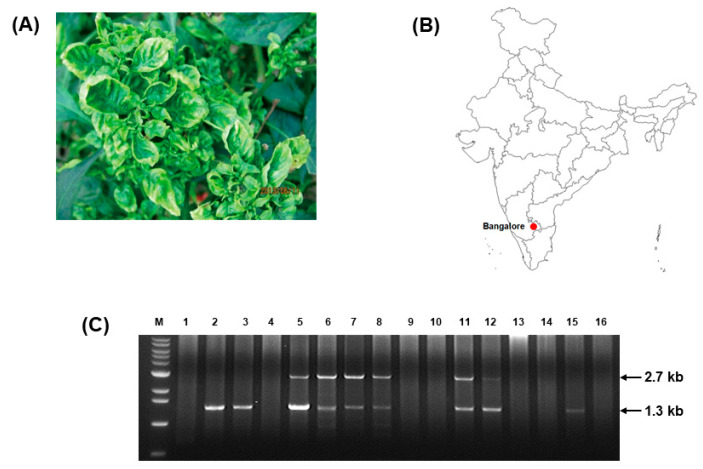
ToLCJoV detection from pepper (*C. annuum*) in India. (**A**) Symptomatic pepper plants collected from Bangalore, Karnataka, India. (**B**) Location of sample collection on map. (**C**) Gel electrophoresis of RCA product digestion with *Bam*HI showed an association of the geminivirus with the expected satellite molecules. Lane M: 100 bp ladder, line 1–16: RCA products of 16 samples after digested by enzyme.

**Figure 2 microorganisms-11-02907-f002:**
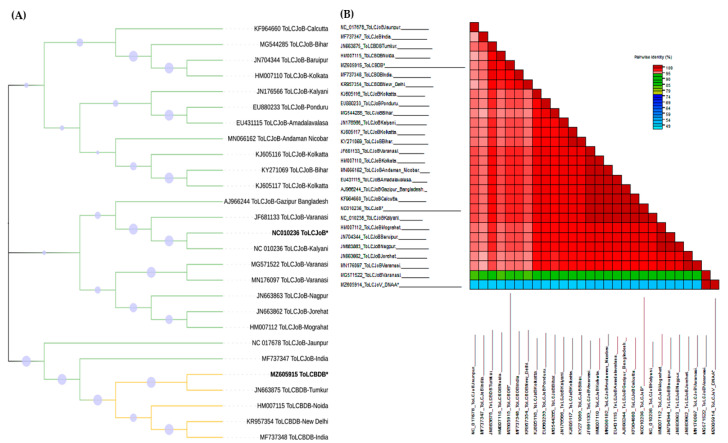
Phylogenetic and SDT analysis of ToLCJoB and ToLCBDB. “*”: the betasatellite isolates used in this study and others available in GenBank were used to generate the phylogenetic tree using the Tamura-Nei model and neighbor-joining method along with 1000 bootstraps: (**A**) relation between ToLCJoB and ToLCBDB through the phylogenetic tree, (**B**) relation of the IR between all viruses that have no association in this study using the SDT.

**Figure 3 microorganisms-11-02907-f003:**
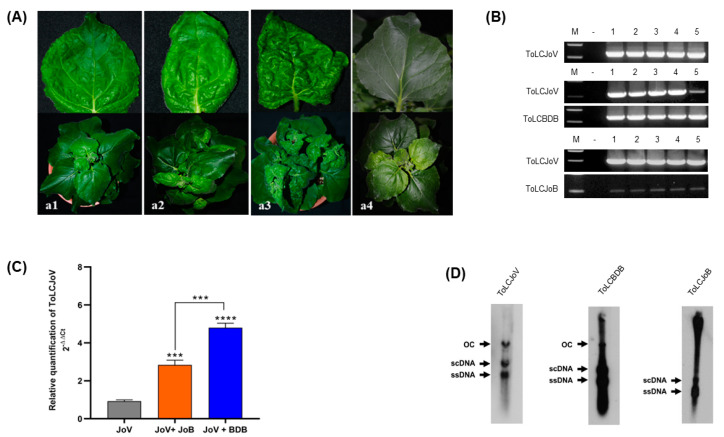
Symptom development and molecular detection in the combination of ToLCJoV and two betasatellites (**A**) Infectivity of ToLCJoV and betasatellites in *N. benthamiana* at 28 dpi: (**a1**) ToLCJoV alone, (**a2**) ToLCJoV × ToLCJoB, (**a3**) ToLCJoV × ToLCBDB, (**a4**) mock plants. (**B**) PCR results in every virus in combination; lane M: 100 bp marker, lane N: negative control, lane 1–5: inoculated plants. (**C**) Relativequantification of ToLCJoV (2^−ΔΔCt^) in the infected plants shown as relative fold change of viral DNAunder different combinations. The bar graph indicates the mean ± standard deviation (n = 3). The statistical comparison was performed using the unpaired *t*-test: *** *p* < 0.001, **** *p* < 0.0001. (**D**) Southern blot hybridization of ToLCJoV, ToLCBDB and ToLCJoB in the infected plants. ssDNA: single-stranded DNA, scDNA: supercoil form, and OC: open circular form of double-stranded DNA.

**Figure 4 microorganisms-11-02907-f004:**
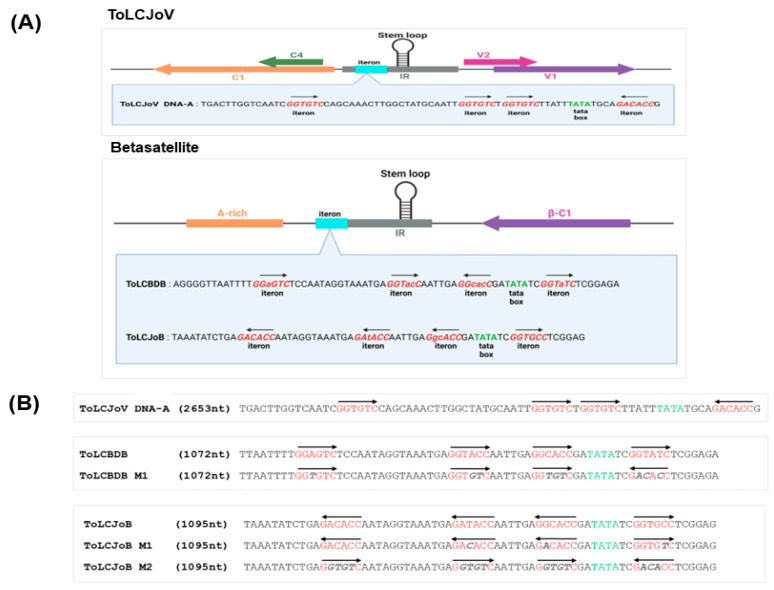
*Iteron* analysis and mutant virus design. (**A**) *Iteron* resemblance analysis of ToLCJoV, ToLCBDB, and ToLCJoB. (**B**) *Iteron* modification for mutant betasatellites of ToLCBDB and ToLCJoB. *Iteron* modification sequence is shown in italic gray letters, and the arrow explains the sequence direction.

**Figure 5 microorganisms-11-02907-f005:**
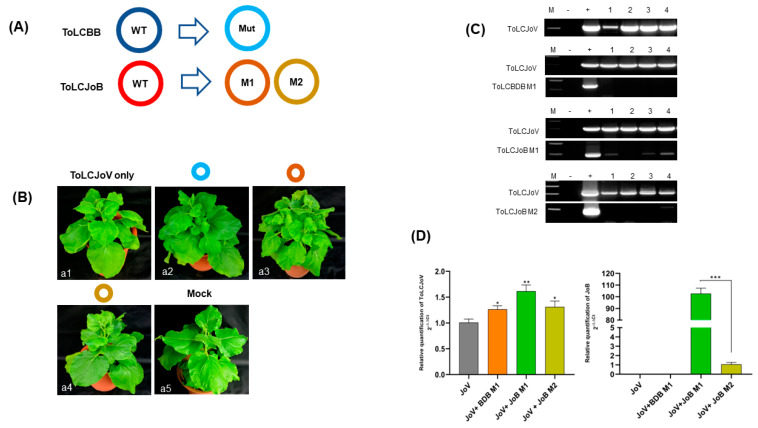
Infectivity assay of ToLCBB and ToLCJoB mutant clones in *N.benthamiana* (**A**) Illustration of mutant genome of each satellite (**B**) Phenotype of mutant betasatelliteagroinoculation combination: (**a1**) ToLCJoV only, (**a2**) ToLCJoV × ToLCBDB M1, (**a3**) ToLCJoV × ToLCJoB M1, (**a4**) ToLCJoV × ToLCJoB M2, (**a5**) Mock plant. (**C**) PCR result of 28-days post inoculation of mutant betasatelliteagroinoculation combination. Lane M: 100 bp marker, lane −: negative control, lane +: positive control, lane 1–4: inoculated plants (**D**) qPCR results to quantify relative fold change of viral DNAfoToLCJoV and ToLCJoB in mutant-clone-inoculated plants.The statistical comparison was performed using the unpaired *t*-test: * *p* < 0.1, ** *p* < 0.01, *** *p* < 0.001.

**Table 1 microorganisms-11-02907-t001:** Primer sets used in this study.

Purpose	Primer	Sequence (5′-3′)	Target Size
Detection	ToLCJoV-det-F	GAAGCCCAGATGTTCCTAGG	554 bp
	ToLCJoV-det-R	GCATACACAGGGTTAGAGGC	
	ToLCBDB-F	TGACCTTGTCGAGCACAATTC	677 bp
	ToLCBDB-R	GATACCGATATATCGGTGCC	
	ToLCJoB-F	CTGGTGACCGTGTGGATAC	489 bp
	ToLCJoB-R	GTGTGTGCTTGTTGTTGTCAG	
Quantitative PCR	qJoV-F	ATTGGGTCCTGGATTGCAGA	143 bp
	qJoV-R	ATGACGTCGATCCCCACTAC	
	qJoB-F	GCCTCTACCATGTCCTCCTG	116 bp
	qJoB-R	GGGATCATACCACCGTTCGA	

**Table 2 microorganisms-11-02907-t002:** Infectivity of different combination in this study.

Virus	Infectivity *		Days of Symptom Appearance	Symptom
	DNA A	Satellite		
ToLCJoV	10/10	-	21	Mild leaf crumpling
ToLCJoV × ToLCBDB	10/10	10/10	14	Severe leaf curling, leaf crumpling
ToLCJoV × ToLCJoB	10/10	10/10	18	Mild leaf curling
ToLCJoV × ToLCBDB M1	10/10	0/10	21	Mild leaf crumpling
ToLCJoV × ToLCJoB M1	10/10	6/10	18	Leaf crumpling
ToLCJoV × ToLCJoB M2	10/10	0/10	21	Mild leaf crumpling

*: number of infected plants/number of inoculated plants.

## Data Availability

Data are contained within the article or Supplementary Material.

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
