# Peer review of "Transreplication Preference of the Tomato Leaf Curl Joydebpur Virus for a Noncognate Betasatellite through Iteron Resemblance on Nicotiana bethamiana"

_microorganisms, 2023, doi:10.3390/microorganisms11122907_

Round 1
Reviewer 1 Report
Comments and Suggestions for Authors
Reviewer 2 Report
Comments and Suggestions for Authors
The presented work is sufficiently equipped with experimental data which is eligible to be used for further understanding of the interaction between monopartite begomoviruses and their associated DNA satellites. It also particularly emphasizes the iteron sequence’s role and direction.
However, minor typos and inaccuracies are still present and are listed below:
Line 62 – Missing space.
Line 69 – CR presented as an undeciphered abbreviation (common region).
Figure 1 – Illustration «B» is irrelevant and should be necessarily included.
Figure 3 – In illustration «B» negative control symbol isn’t the same as in the text below.
Line 306 – Missing space.

Author Response
Line 62 – Missing space.
- The space has been added in the main text.
Line 69 – CR presented as an undeciphered abbreviation (common region).
- The full name of CR has been mentioned in the text.
Figure 1 – Illustration «B» is irrelevant and should be necessarily included.
- In this figure, we just want to indicate exactly the place where we collected samples
Figure 3 – In illustration «B» negative control symbol isn’t the same as in the text below.
- The Negative control symbol in the text has been changed to be same as Figure 3.
Line 306 – Missing space.
- The space has been added in the main text.

Round 2
Reviewer 1 Report
Comments and Suggestions for Authors
L281; recombination analysis is not performed in the revised version
Fig 5; Detection of betasatellite in mock plant
L300: Demonstration of symptoms
Additionally, there are some more problems as;
Fig 1; which marker was used?
L436-437; Sentence looks incomplete
Comments on the Quality of English LanguageMy expertise might be limited, but there's room for improvement in the English quality of certain sentences.
Author Response
L281; recombination analysis is not performed in the revised version
- The recombination analysis has been mentioned in the main text for viral detection part.
Fig 5; Detection of betasatellite in mock plant
- The betasatellite was not detected in mock plants. As we explained in previous revised version, the mock plants got contamination because of our our careless after we rechecked the data. Then we processed new qPCR with the same mock plants that we kept in -80degree freezer, and this time no betasatellite was found.
L300: Demonstration of symptoms
- The symptoms of mutant clone infected plants have been described in the main text.
Fig 1; which marker was used?
- We used 100 bp ladder marker and mentioned it in the Figure 1 legend.
L436-437; Sentence looks incomplete
- According to line number of the latest revised manuscript, the line 436-437 is placed at beginning of Supplementary material part and we did not see the incomplete sentences. We just confused there is mistake in the line number that reviewer mentioned.
